# Significant Preoperative Anxiety Associated with Perceived Risk and Gender in Cataract Surgery

**DOI:** 10.3390/jcm13175317

**Published:** 2024-09-08

**Authors:** Georgios Floros, Stylianos Kandarakis, Nikolaos Glynatsis, Filaretos Glynatsis, Ioanna Mylona

**Affiliations:** 12nd Department of Psychiatry, Aristotle University of Thessaloniki, 54124 Thessaloniki, Greece; 21st Department of Ophthalmology, General Hospital “G. Gennimatas”, National and Kapodistrian University of Athens, 11527 Athens, Greece; s.kandarakis@gmail.com; 3Department of Ophthalmology, Hippokration General Hospital of Thessaloniki, 54642 Thessaloniki, Greece; nmglynatsis@gmail.com (N.G.); fglynatsis@gmail.com (F.G.); 4Department of Ophthalmology, General Hospital of Serres, 62100 Serres, Greece; milona_ioanna@windowslive.com

**Keywords:** anxiety, cataract, perioperative risk factors

## Abstract

**Background/Objectives**: Cataract surgery is an often-sought solution to the universal problem of lens opacification. Studies of perioperative anxiety have yielded conflicting results, reporting a high prevalence but low clinical significance. The objective of this study was to ascertain anxiety levels immediately after the scheduling of surgery, controlling for trait anxiety and other related factors. **Methods**: This is an observational comparative study of two patient populations assessed for receiving cataract surgery, with one group of seventy patients scheduled for operation and receiving an assessment of the potential perioperative risk and the other group of seventy patients deemed ineligible for operation since their opacification was not advanced. The patients were assessed for state and trait anxiety while controlling for cognitive status. **Results**: The findings indicate a clinically significant burden of state anxiety in the group of patients scheduled for operation, with 34 out of 70 meeting the threshold for clinically significant levels of state anxiety compared to 9 out of the 70 patients who were not assigned for surgery (*p* < 0.001). Those patients who were assigned for surgery were assessed for perioperative risk factors, and state anxiety differed statistically significantly between the preoperative risk factor groups, (*p* = 0.003) with those assessed as having at least low perioperative risk presenting with more anxiety than those without any risk factors. Male patients exhibited lower state anxiety compared to female patients in the group assigned to surgery (*p* = 0.028). Cognitive status did not affect the results. **Conclusions**: These findings point to the importance of prevention against perioperative anxiety early on, especially in patients with a higher perioperative risk and female gender.

## 1. Introduction

Opacification of the crystalline lens (‘cataract’) is common in people over the age of 65, with 95% having some degree of lens opacification. A significant proportion have cataracts thick enough to warrant extraction, with rates as high as 38.8% of men and 45.9% of women over the age of 74 [1]. Cataract surgery is the only solution that can restore the patient’s functionality and quality of life to previous levels.

There is a lack of comprehensive research in the field of preoperative anxiety in cataract surgery, leading to conflicting findings. An early study by Foggitt et al. concluded that the typical patient does not exhibit significant anxiety regarding upcoming cataract surgery [2], a finding repeated in a study carried out in the Netherlands [3]. In contrast, a more recent study by Socea et al. revealed that one fifth of the patients who participated experienced severe anxiety and pain [4].

According to Spielberg’s theory [5], anxiety is a single construct that encompasses both state and trait anxiety; these are considered to be different aspects of the same phenomenon. Anxiety can be categorized as either “state anxiety”, a temporary response to stressful events, or “trait anxiety”, which is a more enduring personal characteristic. Trait anxiety is seen as a consistent feature of an individual’s personality and is linked to a tendency to react with worries and concerns in various situations. It is associated with persistent high arousal and certain psychological conditions. On the other hand, state anxiety is a brief, intense emotional state, associated with a temporary increase in the activity of the sympathetic nervous system, but not with any specific psychological conditions. The construct validity of trait anxiety was confirmed with the recent discovery of a neuroanatomical and functional distinction between state and trait anxiety [6]. MRI testing across anxiety types found several differences in the structural–functional patterns regarding structural grey matter and resting-state functional connectivity. Furthermore, in a separate neurophysiological study [7], healthy individuals with high trait anxiety levels were found to be susceptible to anxiety-related psychopathology.

Trait and state anxiety have been found to be neuroanatomically and functionally distinct [6], justifying earlier calls to consider the construct of anxiety as multidimensional [8], contrary to the initial concept of trait and state anxiety as two sides of the same coin. While state anxiety is expected to rise before, during and after cataract surgery, high trait anxiety could be a risk factor for higher state anxiety and needs to be controlled for, since higher trait anxiety is associated with a heightened anxiety response compared to that in lower trait anxiety individuals.

Gender is an important variable since there are well-reported gender differences with regard to both trait anxiety and state anxiety in different settings [9,10]. With regard to conditions of clear and present threat, females may react with more state anxiety than males; in a recent study regarding anxiety during the COVID-19 epidemic [11], female participants had significantly higher mean state anxiety scores than the male participants. Also, a study in the Netherlands found that women with higher trait anxiety are more likely to experience higher levels of state anxiety, although the mean level of anxiety was low [6]. Thus, these complex effects needed to be controlled for. Age and cognitive status are important confounder variables, since cataract tends to progress exponentially with age while old age may be accompanied by varying degrees of cognitive decline. While anxiety is generally associated with a negative effect on cognition, a related study on elders concluded that state anxiety in older adults is not necessarily deleterious to cognitive performance and may even be beneficial [12]. Anxiety tends to peak in cases of full-blown dementia and gradually subsides as the disease progresses [13].

The high prevalence of cataract in old age and its negative impact on patients’ daily life translate to a significant burden of disease globally, with older individuals, females, and those with a lower socioeconomic status associated with a higher cataract burden [14]. The expected improvement in the quality of life is an important motive towards surgical intervention, yet pre-operative anxiety can cause significant mental distress. High anxiety preoperatively has been linked to various adverse surgery outcomes, including severe post-operative pain [15,16], increased morbidity [17] and a poorer sleep quality [16]. Hence, it is important to clarify the prevalence of pre-operative anxiety, in order to develop targeted interventions that are aimed at averting anxiety-related complications and ensuring the early identification of any severe cases.

The primary aim of this study is the comparison of state anxiety between the two groups, controlling for patients’ cognitive ability. The secondary outcomes are the analysis of whether state anxiety can be predicted by the patient’s trait anxiety, whether the patient’s cognitive ability influences state anxiety and whether the genders respond differently in this setting. The primary working hypothesis is that patients scheduled for cataract surgery will demonstrate a clinically significant level of anxiety, which will be statistically significantly higher than the level of anxiety in those patients who will not be deemed eligible for surgery yet and scheduled for re-evaluation in the future. A secondary hypothesis is that pre-existing trait anxiety will be a significant predictive factor for the current level of state anxiety in the patients eligible for surgery. We also predict that the cognitive status of the patient will not play a significant role in the levels of anxiety, but that gender will.

## 2. Materials and Methods

### 2.1. Study Design and Population

This is an observational study with two patient cohorts examined comparatively. The first group consists of patients who were selected for cataract surgery while the second group consists of patients who were deemed to be inappropriate for surgery. The selection criteria for inclusion were patients of any age, those suffering from simple cataract, and those without any other ocular pathologies that may affect visual function. The patient groups were matched for gender by including consecutive patients at a 1:1 gender ratio in each group. Patients were assessed for best-corrected visual acuity (BCVA) in the cataractous eye, BCVA in the contralateral eye, visual function, and the potential surgical complexity of the cataract procedure. Thus, any patient selected for inclusion would have to suffer from a clinically significant reduction in visual acuity that was leading to difficulty with recreational activities or activities of the daily life due to a considerable drop in BCVA in the cataractous eye and/or a concurrent drop in BCVA in the contralateral eye for any reason. The potential complexity of the procedure (leading to a possibly poorer improvement post-operationally) would weigh in if the previous parameters were significant but the patient himself/herself wished to prolong the interval leading to surgery to a later date, after being informed of the potential risk associated with a partial improvement. A separate exclusion criterion for participation in the study was the prescription of psychiatric medication during the current time period for any reason. The control group comprised patients who were deemed inappropriate for the immediate programming of cataract surgery due to a partial loss of visual function that may have been due to a small drop in the BCVA of the affected eye and/or the adequate BCVA of the contralateral eye.

Patients were recruited upon the completion of their assessment for receiving surgery in the outpatient services of the Ophthalmological department of Serres General Hospital. All patients gave their informed consent upon completing the test battery. The patients were sequential; there was an effort to recruit equal numbers of men and women in each research group, and this was nearly achieved. The surgery-assigned group comprised 35 males and 35 females, while the group of patients who were not assigned to surgery (referred as ‘control group’) comprised 36 males and 34 females. The age characteristics are presented in Table 1.

This study was conducted in accordance with the Declaration of Helsinki, and approved by the Institutional Review Board of the General Hospital of Serres (protocol code 32 and date of approval 5 September 2023.

### 2.2. Measures

Trait (structural) and state (situational) anxiety are assessed with the State–Trait Anxiety Inventory (STAI). The STAI (State–Trait Anxiety Inventory) consists of forty statements that describe feelings and are rated, from 1 to 4, in terms of agreement with them, on a 4-point Likert scale (“Not at all” to “Very much”). There are two available scores obtained from the sum of the answers, with a range of 20 to 80. The higher the score, the greater the anxiety the person feels. Originally designed to measure anxiety in a healthy population, but later used as a research tool in the examination of anxiety disorders [18], its derived Greek version was employed in the study [19]. The State Anxiety Scale (S-Al), which measures people’s sentiments at a certain moment in time, has 20 items, 10 of which describe negative emotions and 10 of which represent good emotions. The Trait Anxiety Inventory (T-AI), which has items 21 through 40, is used to measure people’s habitual anxiety experiences and identify the aspects of personality associated with anxiety. Eleven of the items represent negative feelings, and nine reflect positive emotions. Every item is scored using a 4-point system. Higher scores correspond to higher levels of anxiety. The State score for healthy subjects was 34.30 ± 10.79 and the Trait score was 36.07 ± 10.47.

The preoperative risk assessment for the group that was assigned to surgery was carried out using the risk stratification system detailed in a study by Mylona et al. [20]. This system assigns each patient to one of four risk groups, namely the no-risk group, the low-risk group, the moderate-risk group and the high-risk group, depending on the presence of pre-existing risk factors; these are scored depending on their severity and total number. The categorization is carried out by adding up the points for each variable; no risk implies that none of the risk factors are present. Assigning a patient to the low-risk group requires the presence of a single risk factor weighted with one point. The moderate-risk group has a threshold of three points (either three risk factors from the low-risk group or one risk factor from the moderate-risk group). The high-risk group has a threshold of nine points (either three risk factors from the moderate-risk group or one risk factor from the high-risk group).

The screening of cognitive functions was carried out with the MMSE (Mini Mental State Examination), considering a drop in cognition as a confounding factor. The Mini Mental State Examination (MMSE) is a diagnostic tool for assessing a person’s cognitive abilities. It includes thirty items used to test cognitive abilities (problems in thinking, communication, understanding and memory), and its derived Greek version was employed in this study [21]. The maximum score is 30. A score of 23 or lower is indicative of cognitive impairment. The MMSE is a brief test that typically requires 5–10 min to administer and is therefore practical to use repeatedly and routinely.

### 2.3. Statistical Analysis

The descriptive statistics for the measured variables included mean and standard deviation (SD), while the inferential statistics included a univariate analysis of variance in which state anxiety was treated as a dependent variable and gender, trait anxiety and the MMSE total score were treated as independent variables. There was an added effect for the combination of gender and group in order to determine whether the genders responded differently with regard to state anxiety in the setting. This analysis would answer the primary and secondary aims of this study.

## 3. Results

### Results of the STAI and MMSE Test Questionnaires

Table 1 presents the results from the test questionnaires (STAI state anxiety, STAI trait anxiety and MMSE total score) by patient group and gender.

The examination of the first primary hypothesis is as follows: The results from the STAI questionnaire were assessed with the aid of the cut-off values from the Greek normative sample, with values higher than mean + 1SD denoting clinically significant levels of state and trait anxiety. In the surgery-assigned group, there were 34 patients surpassing the state anxiety cut-off value but nine patients in the control group, a statistically significant difference (chi-square = 23.874, *p* < 0.001). There were two patients surpassing the trait anxiety threshold in the surgery-assigned group but none in the control group, a non-significant difference (corrected chi-square = 0.507, *p* = 0.376).

The surgery-assigned group was assessed for preoperative risk factors. In total, 22 out of 70 patients (31.4%) had no risk factors, 31/70 (44.28%) were classified as low-risk patients, 11/70 (15.71%) were classified as moderate-risk patients and 6/70 (8.5%) were classified as high-risk patients. Table 2 presents the STAI state anxiety scores of all risk groups.

An analysis of variance was carried out to determine whether there were any statistically significant differences between the risk groups. The results indicated that state anxiety differed statistically significantly between the preoperative risk factor groups, F(3, 69) = 5.166, *p* = 0.003. The group differences that were statistically significant were those of the no-risk group with the low (*p* = 0.026) and moderate risk (*p* = 0.007) groups. Figure 1 presents the mean plots for the analysis.

The examination of the secondary hypothesis is as follows: A univariate analysis of variance was carried out with STAI state anxiety as the dependent variable and group membership, gender, STAI trait anxiety and the MMSE total score as the independent variables; this was to determine how the independent variables affected the state anxiety levels. Levene’s test for the equality of error variances was non-significant, F(3, 136) = 0.333, *p* = 0.802, indicating that the assumption of homogeneity of variances is not violated.

The results indicated that there was a statistically significant difference between the two groups with regard to state anxiety while taking into account gender, age, trait anxiety and the MMSE scores (F (1, 134) = 150.64, *p* < 0.001, partial eta squared (η^2^) = 0.529, observed power = 1). STAI trait anxiety had a statistically significant and clinically important impact on the STAI state anxiety scores (*p* < 0.001, partial eta squared η^2^ = 0.714 corresponding to an effect size of 0.844, which is very large). Gender by itself did not statistically significantly affect the STAI trait anxiety score (*p* = 0.936); however, there was a statistically significant difference with regard to how the genders were affected by the group condition, with males exhibiting less state anxiety than females in the group assigned to surgery (*p* = 0.028). The partial eta squared η^2^ = 0.035 corresponds to an effect size of 0.187, which is low to moderate. The MMSE values did not statistically significantly impact the STAI state anxiety scores (*p* = 0.33, partial eta squared η^2^ = 0.005). Table 3 and Table 4 present the results from the tests of between-subject effects and the parameter estimates, respectively.

Figure 2 presents the estimated marginal means of STAI state anxiety among the two research groups divided by gender, with error bars denoting 95% confidence intervals. There is a higher relative increase in state anxiety in the females in the cataract-assigned group compared to males, while the trend is reversed in the control group.

## 4. Discussion

The findings from this study point to a significant anxiety burden on patients who are directed to cataract surgery versus patients who are deemed still ineligible for surgery. This is the first study to compare these two populations; those patients deemed eligible for surgery faced the uncertainties of an operation, while those who were not eligible did not have to worry about imminent surgery in the present time, although they should undergo a new assessment in the future. Contrary to previous research [2,3], the levels of state anxiety were considerable in the cataract-assigned group, with nearly half of the patients (34/70) reaching clinically significant anxiety, while trait anxiety was broadly similar among the two groups. Also of note is the fact that even those patients not assigned to surgery presented with higher mean values of state anxiety than normal, although the number of cases of high anxiety was still low (2/70). Thus, we should not negate the anxiety of those patients waiting for an eventual operative procedure some time down the line.

State anxiety was markedly higher when the risk of preoperative complications was assessed as higher. This is also a novel finding. Patients are typically informed about the possibility of complications to reduce expectations and avoid any potential litigation. However, this probably had a significant impact on their anxiety levels, as measured after they had received the relevant information.

The levels of anxiety in patients who were deemed eligible for surgery and patients with preoperative risk factors, in particular, point to the importance of measures taken to reduce potential anxiety as early as during the selection process for surgery. Cataract surgery is a minimally invasive procedure undertaken without general anesthesia. As such, it is devoid of the risk associated with larger and more complex operations. However, undertaking a procedure while fully conscious has its own challenges, and the patient may be reluctant to admit any fears or have difficulty phrasing them due to advanced age. In a related study that focused on the patient’s point-of-view regarding cataract surgery, older patients have been found to have a lesser understanding of the specific stages of the surgery, with the authors suggesting that they should be notified in advance about all the routine procedures to be carried out and provided with a rationale for these activities [22]. When comparing patients who had had prior cataract surgery in the contralateral eye with patients who were having their first surgery [23], those with prior experience had less anxiety, pointing to the simple fact that prior experience alleviated fears. Yet, patients from both groups did not understand the medical information provided by doctors regarding the surgery and its possible complications. Those data show the importance of spending time with the patient to convey the situation and the details of the proposed procedure to amend this situation. While cataract surgery may be routine for the experienced surgeon, it is a unique experience for the patient. Various interventions have been suggested to combat preoperative anxiety in cataract surgery, although none have reached mainstream acceptance. A 2013 study detailed how an intervention group of 84 patients who received an audio CD containing information, relaxation, and positive imagery was calmer before and more cooperative during surgery, although there was no difference in sleep quality, subjective well-being or heart rate during the operation [24]. A similar intervention carried out with only the provision of an education manual versus the provision of no material also showed promising results, although the effect sizes were small [25]. Morrell [26] tested the relative efficacy of providing a lengthy self-education information packet for the patient to study alone versus having a registered nurse supervise the viewing of educational videos in the clinical facility and go through questions with them; the latter was found to be superior in all measures of state anxiety, including STAI levels and neurophysiological variables. Gong et al. [27] carried out a study on the effects of providing patients scheduled for cataract surgery with a structured nursing intervention that included presentations on the physiology of the eye, cooperation during surgery, anxiety relief, pain management and rehabilitation care. This intervention helped to increase satisfaction and cooperation, and decrease anxiety.

No comparison among patients with different levels of preoperative risk and the impact on their anxiety levels has been published, but it is reasonable to expect that this would be a significant factor in increasing anxiety, particularly since the risk odds are poorly translated into layman terms. Cataract surgery is an elective surgery where the associated risk is relevant to the success of the procedure and not to the general health of the patient. Also, there is no alternative to surgery for the patient to regain his/her visual function. Hence, the patient may feel that there are few options left for them, regardless of whether the surgery does or does not succeed, since there is no alterative and his/her general health will not be jeopardized regardless of the outcome. This essentially removes the patients’ sense of agency from a decision that is nominally theirs, and they may feel forced into taking it.

Trait anxiety was a statistically significant predictor of state anxiety in our sample, with a large effect size. This finding is in line with previous research that found that patients with high trait anxiety demonstrated higher-than-average state anxiety in a variety of clinical settings, including cataract surgery [28]. Individuals with high trait anxiety are susceptible to developing anxiety disorders or depression brought on by stress because they exhibit hyper-reactivity to stressful events, heightened passive coping responses to environmental challenges, altered cognitive functions, and decreased social competitiveness [29]. State and trait anxiety shared common topological mechanisms of human brain networks in related studies [30,31]. When faced with unexpected anxiety-related events, people with high trait anxiety would get more and more nervous, which would raise their levels of state anxiety. Participants with low trait anxiety, on the other hand, would react to threats with a resistant and defensive response, which would lower their state anxiety levels. The results from our study confirm this general hypothesis in a clinical setting.

Males reacted with less state anxiety in the surgery-assigned group than females, results that mirror those of other studies where the patient was facing a potential threat [11]. Women, relative to men, reported greater panic symptoms and demonstrated increased startle potentiation in anticipation of predictable and unpredictable threat [32]. Cataract surgery, although limited in its scope compared to other surgery, may still evoke a response to threat in the patient. Similar results were found in the study by Nijkamp et al. [3], although that study did not include a control condition, the anxiety levels were low and the statistical exploration was limited to correlation. There is little research regarding anxiety levels in other ophthalmic diseases; a study of patients receiving intravitreal injections [33] did not reveal any differences either in state or trait anxiety between the genders, despite patients presenting with significant anxiety levels.

When examining the age difference between the groups, the surgery-assigned group was older, although the difference was not statistically significant. Age did not affect state anxiety in the multivariate analysis. Although state anxiety tends to increase with age, this difference eventually reaches a plateau and both groups had relatively high mean ages, so any effect was minimized.

There was no significant difference in the cognitive status of the two groups, as measured by the MMSE. While cataract surgery has been effective in decreasing the risk of fall, improving depressive symptoms and limiting the risk of cognitive impairment and dementia [34], patients’ cognitive status pre-surgery did not affect state anxiety levels. This confirms previous research that, outside the large cognitive deficits that are found in cases of dementia, there is little correlation between cognitive status and state anxiety [12].

Finally, this patient sample was not receiving medication for any mental health issue. It is reasonable to expect that the dosages of any anxiolytic medication that patients could be receiving would have to be adjusted following a consultation with the attending psychiatrist immediately after being scheduled for surgery.

There are a number of limitations that may reduce the generalizability of these findings. The principal limitation of this study is its cross-sectional nature, which only measures anxiety immediately after the classification of the patients and risk assessment, and that it does not follow through until surgery. It is thus unclear how long the high levels of state anxiety persist and what may help the patients tackle them. The size of the research samples was limited to seventy patients each; however, the results from the multivariate analysis indicated that the study was adequately powered. Also, any random recent adverse life events as possible sources of anxiety were not recorded; however, there is no reason to hypothesize that those would favor one research group over the other.

## 5. Conclusions

This study examined the difference in anxiety levels of patients scheduled for cataract surgery versus patients who were told that it could be postponed to a later date. Patients who were scheduled for surgery had considerably higher anxiety and this was more pronounced in those patients who were told that their surgery had more preoperative risk. Higher levels of trait anxiety were linked with more state anxiety, while male patients responded better than female patients. These findings are important for anxiety prevention among cataract patients since they points to high anxiety very early on in the processing of those patients. A clinical implication is that the patients should be screened early for anxiety related to upcoming surgery, since it could lead to significant anguish and affect decision-making. Liaison psychiatry services would be helpful in addressing clinically significant cases of anxiety. Also, patients may not be able to appropriately assess any information regarding the potential preoperative risk leading to more anxiety in those patients with higher risk. An explanation of the concept of relative risk in lay terms could be helpful in a clinical setting so as not to unnecessarily raise anxiety. Previous research demonstrated the efficacy of patient information in reducing anxiety and improving outcomes in patients scheduled to receive cataract surgery, especially when specialized staff (registered nurses) are supervising the provision of patient education.

## Figures and Tables

**Figure 1 jcm-13-05317-f001:**
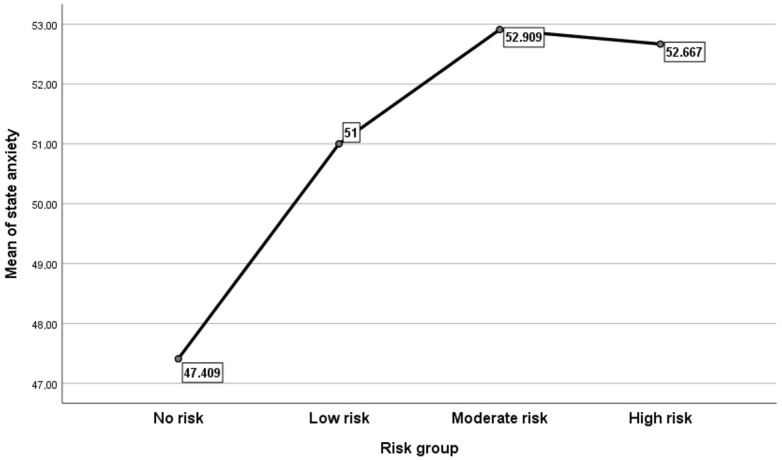
Means plots for the ANOVA test for the differences in state anxiety between the risk factor categories of the patients scheduled for operation.

**Figure 2 jcm-13-05317-f002:**
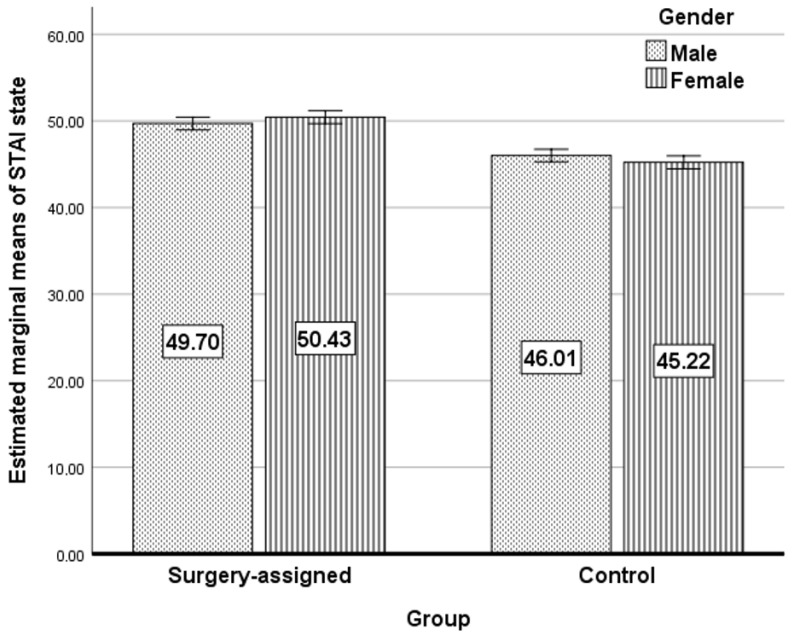
Estimated marginal means of STAI state anxiety among the two research groups.

**Table 1 jcm-13-05317-t001:** Age and results of the STAI and MMSE test questionnaires by group and gender.

		Age		STAI State		STAI Trait		MMSE Total	
Group		Mean	SD	Mean	SD	Mean	SD	Mean	SD
Surgery	Male (35)	66.94	2.96	49.028	4.724	36.085	4.441	28.285	2.051
	Female (35)	67.68	3.14	51.60	4.672	38.20	4.745	27.714	2.051
	Total	67.31	3.05	50.314	4.841	37.142	4.685	28.00	2.057
No surgery	Male (36)	66.55	4.11	45.25	3.324	36.027	3.238	28.194	1.864
	Female (34)	65.79	3.17	45.50	3.350	37.352	3.264	27.588	2.076
	Total	66.18	3.68	45.371	3.315	36.671	3.295	27.90	1.979
Total	Male (71)	66.74	3.57	47.112	4.470	36.056	3.850	28.239	1.945
	Female (69)	66.75	3.27	48.594	5.079	37.782	4.076	27.652	2.049
	Total	66.75	3.41	47.842	4.82	36.907	4.042	27.95	2.011

SD = standard deviation, STAI state = state anxiety measure, STAI trait = trait anxiety measure.

**Table 2 jcm-13-05317-t002:** State anxiety scores between the preoperative risk factor groups.

Risk Factor Groups	N	Mean	Std. Deviation
No risk	22	47,409	4.66
Low risk	31	51	4.04
Moderate risk	11	52.91	4.06
High risk	6	52.66	6.28
Total	70	50.31	4.84

**Table 3 jcm-13-05317-t003:** Multivariate analysis for STAI state anxiety: tests of between-subject effects.

	F	*p*	Partial Eta Squared
Corrected Model	91.305	<0.001	0.805
Intercept	2.460	0.119	0.018
Group	140.852	<0.001	0.514
Gender	0.006	0.936	<0.001
Group * Gender	4.905	0.028	0.035
STAI Trait anxiety	338.247	<0.001	0.718
Age	1.698	0.195	0.007
MMSE total	0.956	0.330	0.005

**Table 4 jcm-13-05317-t004:** Multivariate analysis for STAI state anxiety: parameter estimates.

Parameter	B	Std. Error	t	*p*	95% Confidence Interval		Partial Eta Squared
					Lower Bound	Upper Bound	
Intercept	5.656	5.232	1.081	0.282	−4.693	16.004	0.009
Group = 1	5.213	0.538	10.160	<0.001	4.149	6.277	0.414
Male gender	0.791	0.530	1.493	0.138	−0.257	1.840	0.016
Group = 1 * Male gender	−1.637	0.739	−2.215	0.028	−3.100	−0.175	0.035
STAI Trait anxiety	0.87	0.047	18.391	<0.001	0.776	0.963	0.718
Age	0.073	0.056	1.303	0.195	−0.038	0.184	0.013
MMSE total	0.074	0.093	0.790	0.431	−0.111	0.258	0.005

* Group 1 = patients assigned to cataract surgery.

## Data Availability

Data are available from the authors upon reasonable request.

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
