# Peer review of "Significant Preoperative Anxiety Associated with Perceived Risk and Gender in Cataract Surgery"

_jcm, 2024, doi:10.3390/jcm13175317_

Round 1
Reviewer 1 Report
Comments and Suggestions for Authors
I have some questions for understanding this paper.
In Table 1. I think the authors add the number of patients in each group. I could not guess the number of patients in each group from the description “2.1. Study design and population” and Figure legends.
Why the authors describe Fig.1? It is the same as Table 2, I think it is unnecessary.
Author Response
In Table 1. I think the authors add the number of patients in each group. I could not guess the number of patients in each group from the description “2.1. Study design and population” and Figure legends.
Response: Thank you for your comment, the Table has been updated accordingly to include patient numbers
Why the authors describe Fig.1? It is the same as Table 2, I think it is unnecessary.
Response: As per the article guidelines each figure needs its own description
Reviewer 2 Report
Comments and Suggestions for Authors
Thank you for the opportunity to review this manuscript. The topic is interesting, but the paper needs some revision.
Abstract
What do the authors mean with “constitutional anxiety”? Please adds more detail on method variables examined and results.
Introduction
The introduction needs to be enriched. particularly with regard to medical aspects, in order to better understand why it would be important to investigate these aspects in patients and what led to the study specifically.
The aims of the study are not stated at the end of the section. Thus, the objectives are not clear, and specific hypotheses tested should also be added.
Method
In the methods, the inclusion and exclusion criteria are not clearly spelled out. Was a specific age range chosen?
The sub-section “2.2. Choice of examined variables.” should be integrated into the introduction and consequently into the aim of the study.
It is important to give clear description of the instrument used.
Data analysis should be well described, in line with objectives and hypothesis.
Results
the tables should indicate the numerosities of the total sample and the subgroups. there is also a lack of notes completing the table by including all the descriptions of the acronyms.
Why did the authors investigate the impact of trait anxiety on state anxiety. What is the hypothesis, and the objective pursued? It is well known that there is a high correlation between these measures, and they should not be used to predict each other. Did the authors make sure that there was no multicollinearity?
It should also be considered to revise the language used, as the data is cross-sectional, and it is not possible to establish strong causal relationships between the variables.
Discussion
The discussion should be enriched with references and restructured to improve clarity and readability.
A paragraph on the limitations of the study is missing.
Finally, the conclusion should be expanded, particularly regarding the clinical implications that the results of the study may have.
Comments on the Quality of English Language
Even though I'm not a native English speaker, I would suggest to the authors that they revise the English by seeking assistance from a native speaker or utilizing a professional linguistic editing service.
Please, carefully proofread the manuscript and fix the many typos present (e.g., in line 34 “a finding repeated in a study in a 2004 study in the Netherlands.”)
Author Response
Thank you for the opportunity to review this manuscript. The topic is interesting, but the paper needs some revision.
Abstract
What do the authors mean with “constitutional anxiety”? Please adds more detail on method variables examined and results.
Response: Thank you for your constructive suggestion, the abstract has been amended accordingly
Introduction
The introduction needs to be enriched. particularly with regard to medical aspects, in order to better understand why it would be important to investigate these aspects in patients and what led to the study specifically.
Response: Thank you for your constructive suggestion, the Introduction section has been amended accordingly
The aims of the study are not stated at the end of the section. Thus, the objectives are not clear, and specific hypotheses tested should also be added.
Response: Thank you for your comment, the aims have been added in the end of the section
Method
In the methods, the inclusion and exclusion criteria are not clearly spelled out. Was a specific age range chosen?
Response:Thank you for your comment, ere was no specific age range chosen and this has been made clear in the revised manuscript. The groups were matched for gender in 1:1 basis.
The sub-section “2.2. Choice of examined variables.” should be integrated into the introduction and consequently into the aim of the study.
Response: Thank you for your helpful suggestion, this has been implemented.
It is important to give clear description of the instrument used.
Response:All three instruments have been described in the section 2.2 and additional information has been included
Data analysis should be well described, in line with objectives and hypothesis.
Response:Thank you for your suggestion, section 2.3 and parts of the results sections have been expanded to help the reader follow the rationale of the study
Results
the tables should indicate the numerosities of the total sample and the subgroups. there is also a lack of notes completing the table by including all the descriptions of the acronyms.
Response:Thank you for your suggestion, changes were made to this effect
Why did the authors investigate the impact of trait anxiety on state anxiety. What is the hypothesis, and the objective pursued? It is well known that there is a high correlation between these measures, and they should not be used to predict each other. Did the authors make sure that there was no multicollinearity?
Response: Thank you for your comment. Numerous studies have included trait anxiety as a control variable in the assessment of state anxiety since higher trait anxiety is associated with a heightened anxiety response compared to lower trait anxiety individuals and this was the purpose here as well. This is based on recent findings that point to the multidimensionality of anxiety; particularly that trait and state anxiety have been found to be neuroanatomically and functionally distinct. Thisexplanation has been added in the manuscript with appropriate citations (page 2, lines 57-62).
It should also be considered to revise the language used, as the data is cross-sectional, and it is not possible to establish strong causal relationships between the variables.
Response: Thank you for your suggestion, changes were made to reflect this throughout the manuscript
Discussion
The discussion should be enriched with references and restructured to improve clarity and readability.
Response: Thank you for your suggestion, the discussion section has been expanded and included your earlier suggestions
A paragraph on the limitations of the study is missing.
Response: Thank you for your suggestion, a paragraph on the limitations has been added in the manuscript.
Finally, the conclusion should be expanded, particularly regarding the clinical implications that the results of the study may have.
Response: Thank you for your suggestion, the section has been expanded.
Comments on the Quality of English Language
Even though I'm not a native English speaker, I would suggest to the authors that they revise the English by seeking assistance from a native speaker or utilizing a professional linguistic editing service.
Please, carefully proofread the manuscript and fix the many typos present (e.g., in line 34 “a finding repeated in a study in a 2004 study in the Netherlands.”)
Response: Thank you for your comment, the entire text has been proofed by a native speaker with numerous changes (highlighted throughout)
Reviewer 3 Report
Comments and Suggestions for Authors
Floros and colleagues tried to ascertain anxiety levels immediately after the scheduling of surgery, controlling for constitutional anxiety and other related factors. This is an observational comparative study of two patient populations assessed for receiving cataract surgery. Findings indicate a clinically significant burden of state anxiety in the group of patients scheduled for operation, which was higher in those patients who were assessed as having perioperative risk.
The study is interesting.
I have the following comments:
1) Study design. Authors means they conducted case-control study. Authors should check it again using the following reference: https://www.ncbi.nlm.nih.gov/pmc/articles/PMC2998589/
I feel, the present study is cohort study, and not case-control study.
2) Abstract. Authors should present the most important results with numbers , not only with words/text.
3) Limitations: authors should describe study limitations in more details including small numbers, stud design, some missing variables and much more
Author Response
Floros and colleagues tried to ascertain anxiety levels immediately after the scheduling of surgery, controlling for constitutional anxiety and other related factors. This is an observational comparative study of two patient populations assessed for receiving cataract surgery. Findings indicate a clinically significant burden of state anxiety in the group of patients scheduled for operation, which was higher in those patients who were assessed as having perioperative risk.
The study is interesting.
I have the following comments:
1) Study design. Authors means they conducted case-control study. Authors should check it again using the following reference: https://www.ncbi.nlm.nih.gov/pmc/articles/PMC2998589/
I feel, the present study is cohort study, and not case-control study.
Response: Thank you for your point, it has been addressed in the revised manuscript
2) Abstract. Authors should present the most important results with numbers , not only with words/text.
Response: Thank you for your point, it has been addressed in the revised ABSTRACT
3) Limitations: authors should describe study limitations in more details including small numbers, stud design, some missing variables and much more
Response: Thank you for your suggestion, a section describing the limitations of the study has been included in the end of the discussion
Round 2
Reviewer 2 Report
Comments and Suggestions for Authors
Thank you to the authors or the work done and to consider my previous comments.
However, some aspects can still be implemented. I include below previous comments that may still be useful in implementing the paper.
The introduction needs to be enriched. particularly with regard to medical aspects, in order to better understand why it would be important to investigate these aspects in patients and what led to the study specifically.
The objectives are not clear, and specific hypotheses tested should also be added.
The discussion should be enriched with references.
Author Response
Thank you to the authors or the work done and to consider my previous comments.
However, some aspects can still be implemented. I include below previous comments that may still be useful in implementing the paper.
Response: Thank you for your considerable effort in reviewing our manuscript, this has led to considerable improvements. Detailed responses to your new comments are below:
Comment 1: The introduction needs to be enriched. particularly with regard to medical aspects, in order to better understand why it would be important to investigate these aspects in patients and what led to the study specifically.
Response: thank you for your constructive suggestion, the introduction section has been enriched following your directions (lines 87-89)
Comment 2: The objectives are not clear, and specific hypotheses tested should also be added.
Response: thank you for your constructive suggestion, specific hypotheses have been added, lines 95 - 101
Comment 3: The discussion should be enriched with references.
Response: thank you for your constructive suggestion, the discussion section has been enriched with a relevant section on providing patient education with a number of references (lines 284-300) and elsewhere (342-346, etc)
Reviewer 3 Report
Comments and Suggestions for Authors
no comments
Author Response
Comments: "no comments"
Response: no response necessary
Thank you for your time and your comments, that have led to a considerable improvement on our manuscript